# Electrical Stimulation in Cartilage Tissue Engineering

**DOI:** 10.3390/bioengineering10040454

**Published:** 2023-04-07

**Authors:** Raminta Vaiciuleviciute, Ilona Uzieliene, Paulius Bernotas, Vitalij Novickij, Aidas Alaburda, Eiva Bernotiene

**Affiliations:** 1Department of Regenerative Medicine, State Research Institute Centre for Innovative Medicine, Santariskiu g. 5, 08410 Vilnius, Lithuania; raminta.vaiciuleviciute@imcentras.lt (R.V.); ilona.uzieliene@imcentras.lt (I.U.); bernotaspaul@gmail.com (P.B.); 2Department of Immunology, State Research Institute Centre for Innovative Medicine, Santariškių g. 5, 08410 Vilnius, Lithuania; vitalij.novickij@imcentras.lt; 3Faculty of Electronics, High Magnetic Field Institute, Vilnius Gediminas Technical University, Plytines g. 27, 10105 Vilnius, Lithuania; 4Life Sciences Center, Institute of Biosciences, Vilnius University, Sauletekio al. 7, 10257 Vilnius, Lithuania; aidas.alaburda@gf.vu.lt; 5VilniusTech, Faculty of Fundamental Sciences, Sauletekio al. 11, 10223 Vilnius, Lithuania

**Keywords:** cartilage, osteoarthritis, chondrogenesis, electrical stimulation, mesenchymal stem cells

## Abstract

Electrical stimulation (ES) has been frequently used in different biomedical applications both in vitro and in vivo. Numerous studies have demonstrated positive effects of ES on cellular functions, including metabolism, proliferation, and differentiation. The application of ES to cartilage tissue for increasing extracellular matrix formation is of interest, as cartilage is not able to restore its lesions owing to its avascular nature and lack of cells. Various ES approaches have been used to stimulate chondrogenic differentiation in chondrocytes and stem cells; however, there is a huge gap in systematizing ES protocols used for chondrogenic differentiation of cells. This review focuses on the application of ES for chondrocyte and mesenchymal stem cell chondrogenesis for cartilage tissue regeneration. The effects of different types of ES on cellular functions and chondrogenic differentiation are reviewed, systematically providing ES protocols and their advantageous effects. Moreover, cartilage 3D modeling using cells in scaffolds/hydrogels under ES are observed, and recommendations on reporting about the use of ES in different studies are provided to ensure adequate consolidation of knowledge in the area of ES. This review brings novel insights into the further application of ES in in vitro studies, which are promising for further cartilage repair techniques.

## 1. Introduction

Electrical stimulation (ES) has attracted a lot of attention as a physical stimulus used for tissue engineering and treatment of various diseases, such as movement, psychiatric and seizure disorders, in order to reduce pain and to improve the quality of life [1,2,3]. ES is frequently used for the stimulation of cells in vitro and in vivo, inducing a number of intracellular pathways involved in the regulation of cell metabolism, proliferation, migration and differentiation [4,5]. A meta-analysis of clinical trials showed that neuromuscular electrical stimulation or interferential current can improve pain management and physical function in knee osteoarthritis patients [6,7].

ES has already been proven as a useful tool in cartilage tissue engineering. Cartilage is composed of only non-excitable cells—chondrocytes [8,9]. Due to a lack of voltage-gated Na^+^ and Ca^2+^ channels, these non-excitable cells cannot generate action potential as a response to membrane depolarization [10]. Since the number of chondrocytes in cartilage is very low and their ability to restore damage to the extracellular matrix (ECM) is very weak, cartilage is prone to the development of degenerative diseases such as osteoarthritis (OA) [11]. Various arthroscopic cartilage intervention procedures such as chondroplasty, microfracture or mosaicplasty [12,13,14], as well as more modern technologies such as autologous chondrocyte implantation [15] are currently under development; however, these methods have not yet been approved for clinical use because their efficacy is still to be confirmed. Stem cell-based tissue engineering technologies, specifically those utilizing adult tissue-derived mesenchymal stem cells (MSCs), which can differentiate into chondrocytes, seem to be a promising therapeutic approach for cartilage damage repair. ES has been shown to be an important part in stem cell-based cartilage engineering, as it stimulates chondrogenic differentiation of MSCs even in the absence of growth factors [16,17].

The lack of standardized protocols for ES in tissue engineering introduces challenges in characterizing the exact mechanisms of its effect, since the electrical parameters (applied voltage, pulse or stimulus duration, frequency and field strength) can vary by several orders of magnitude.

This review summarizes ES applied in chondrogenic differentiation experiments. ES regimens, such as continuous, static, cyclic or pulsed stimulation, are described, emphasizing their beneficial effects and limitations. ES parameters and effects in the context of chondrogenesis are presented in order to improve the consolidation of knowledge in this area and direct the research towards development of optimized and more standardized protocols.

## 2. Electrical Stimulation Overview

ES induces electrically mediated stress in cells and changes their membrane potential, which, depending on the protocol used, may lead to either activation of ion channels and other voltage sensitive proteins, or permeabilization of the plasma membrane. These processes result in a flux of different ions across the plasma membrane [18]. Altered ion concentration leads to the activation of different gene expression [19], production and secretion of growth and transcription factors [16,20], cell adhesion [21] and cell-cell interaction molecules [22]. Typically, low electric fields are used in applying ES, which leads to moderate change in the membrane potential by a tenth of mV and initiates the movement of voltage-sensing domains, resulting in conformational changes and the opening of voltage-gated ion channels (Figure 1). The most important voltage-gated channel for chondrogenic differentiation is L-type voltage-gated calcium channel (VGCC), which regulates expression of chondrogenesis markers (SOX9, COL2A1, Ihh) in vitro and limb development in vivo [23]. However, when the cell is exposed to a high-intensity pulsed electric field (PEF), the cell plasma membrane is polarized and a significant transmembrane potential (TMP) is induced [24,25]. When a critical TMP threshold is reached, which is frequently referred as 1 V [26], hydrophilic pores are formed in the membrane, resulting in increased membrane permeability to exogeneous molecules [27]. This phenomenon, called electroporation or electropermeabilization [28], is used for gene and drug delivery, tissue ablation, protein extraction and food processing. Depending on the PEF parameters, electroporation can be reversible or irreversible [29,30]. In mammalian cells, electroporation can be triggered at a 400–800 V/cm PEF [31], while lower field (<400 V/cm) can induce a phenomenon known as electroendocytosis, which means enhanced absorption of macromolecules after cell exposure to low electric fields [32,33]. Scientific papers focusing on the stimulating effects of high-intensity PEF also have started to appear in recent years.

ES is widely applied in various fields of cellular research. After the analysis of Clarivate Analytics Web of Science using keywords “electrical stimulation” and “cells”, 11,180 papers published since 2008 were filtered. The most dominating keywords are related to electrical aspects (electrode, device, current and amplitude), while for biological aspects—expression, receptor, differentiation and inhibition are most widely used (Figure 2). The dominating keyword “rat” shows the application of ES for in vivo studies with rats as the model organism.

The cell membrane is the first to respond to ES, and is critical in maintaining membrane potential, cellular homeostasis and controlling the exchange of nutrients, waste products and chemical molecules important for signaling [34]. ES activates several independent signal transduction pathways; therefore, it is difficult to establish a direct link between ES and specific cellular responses. It is known that ES activates JNK/CREB-STAT3, ERK/JNK/STAT3 and wnt/β-catenin signaling pathways, leading to enhanced phosphorylation of JNK, CREB and STAT3 [35] and the expression of β-catenin protein [36].

There are hypotheses that electric fields and fluid shear stress activate similar signaling pathways that involve integrin receptors [37]. One of the main cell responses to ES is the opening of voltage-gated calcium channels (VGCC) and subsequent increase of calcium (Ca^2+^) inside the cell [38]. Both chondrocytes and mesenchymal stem cells have VGCC that can be regulated by external chemical or physical stimuli [39,40]. Such increase of intracellular Ca^2+^ was also observed in vitro after mechanical stimulation [41]. Furthermore, cytoskeletal structure reorganization, including denser f-actin texture and aligned actin filament orientation, has been observed in response to ES [42] as well as inverse when mechanical stimulation causes intracellular electrical signals through mechanotransduction [37]. The interconnectedness of the effects of ES and mechanical loading might be used in cartilage tissue engineering as scaffolds that cannot withstand mechanical pressure and could be instead stimulated with electrical fields, causing a similar effect as mechanical loads.

In vitro studies show that ES can increase Ca^2+^-driven ATP oscillations, leading to condensation of cells—the initial step of chondrogenic differentiation [22,43]. Furthermore, ES elevates the secretion of growth factors (transforming growth factor beta 1 (TGF-β1), platelet-derived growth factor (PDGF)-AA, and insulin-like growth factor-binding protein 2 and 3 (IGFBP-2 and 3) [22], which further drive the production of ECM, creating a microenvironment for chondrocyte attachment and interactions [44,45].

ES is a powerful physical stimulus that triggers different cell behaviors; however, due to a lack of a systemic approach to characterize the ES parametric protocols, the effects on different cell cultures and scaffolds are hardly predictable.

## 3. Electromechanics of Articular Cartilage

Cartilage is a biphasic tissue, composed of a solid phase, which consists of a charged porous collagen-proteoglycan matrix, and an interstitial fluid phase [46]. Changes in cartilage composition and arrangement of collagen fibers result in different biomechanical properties. It was shown that resistivity gradually increases from the superficial to the deep zone of cartilage, which means that the superficial zone of cartilage contains more mobile charged particles, than the deeper zones, and conducts electrical charge more efficiently. The elastic modulus also increases going from the superficial to the deep zone while the permeability of cartilage decreases [47]. There is not much data about the conductivity of articular cartilage (Table 1). More studies in this field are needed because part of the data were obtained using animal models, in which possibilities to reflect human data are limited.

Articular cartilage undergoes a number of biomechanical and physiochemical changes related to age, obesity, injury or development of OA. One of the most important pathological changes of cartilage in OA is hypertrophic differentiation and fibrillation, when the tissue loses ECM (proteoglycans and collagen) fibers. Cell senescence induces inflammation, which triggers chondrocytes to produce cytokines and catabolic agents (matrix metalloproteinases (MMPs) and aggrecanases (ADAMTS-4 and ADAMTS-5), leading to the destruction of pericellular and intercellular matrix [50]. Cartilage breakdown products activate synovium, resulting in subsequent production of cytokines and infiltration of immune cells, such as macrophages [51]. Similar secretion of pro-inflammatory cytokines and proteinases can be found in cartilage after joint trauma, which can also be a cause of OA [52].

Based on that, ES has been proposed as a tool in cartilage tissue engineering to improve regenerative, mechanical and other properties of engineered tissue. Basic mathematical models of electrical behavior in cartilage have been described previously [46]. Cartilage itself is an electrically charged tissue that exhibits electromechanical, depth-dependent properties [53]. Exercise and other weight-bearing movements cause mechanical deformation of the tissue, producing electrical signals through the flow of positively charged particles across negatively charged ECM of the cartilage [54]. Age, injury-, or pathology-caused reduction of cartilage mobility can reduce endogenous electrical signals. Therefore, the application of external ES can mimic the endogenous electrical signals of the tissue, produce compression and deformation, which results in tissue recovery upon ES [53,55,56].

It was shown that elevated extracellular calcium (eCa^2+^) concentration inhibits chondrogenesis [57] while eCa^2+^ oscillations trigger Ca^2+^-depending transcription factors and signaling pathways and are associated with modifications in cartilage ECM synthesis [58]. This suggests that different functional responses are determined via specific calcium signaling patterns.

In conclusion, this is why externally applied ES, through the regulation of calcium signaling pathway, is a promising, non-invasive stimuli treating OA. Even though ES studies on cartilage have been carried out in vitro, further investigation is still required.

Therefore, electrical signals should be considered as an important component of cartilage function, and the application of external ES may restore Ca^2+^ homeostasis and cartilage tissue integrity after damage or disease.

## 4. Types of ES Application and Their Effects Activating Cellular Mechanisms and Functions

There are several ways how ES can be delivered to cells. The electric field can be applied to cells using electrodes directly inserted into the culture media or using capacitive or inductive coupling (reviewed in Chen, 2019) [1]. In this review, we focused mainly on the research where direct and capacitive coupling was used. Direct coupling is easy to operate but electrode contact with the medium can cause changes in medium temperature, pH, and the generation of reactive oxygen species. Capacitive coupling is non-invasive; electrodes are outside the plate at opposing ends, providing a relatively uniform electrical field to a cell monolayer on a scaffold [1].

The effects of ES on cellular behavior depend on the electrical field parameters and stimulus protocols used. Such stimulus can be directly applied as straightforward continuous static voltage on tissue culture as well as more complicated stimulation with pulses of various waveforms. The main parameters describing ES pulses are: the strength of the electric field (mV/mm), pulse shape (monophasic or biphasic), and duration frequency (Hz). These pulses can be applied in various regimens, and in case of weak electric fields, the total stimulation duration can last hours or even days. Some stimulation protocols may have a complex on–off cycle structure (for example ES 3 h a day for 21 days). Examples of different parameters and the corresponding effects of ES are described in Figure 3 [59,60,61,62,63,64,65].

ES parameters are determined not just by cell types, but also by different study end-goals and ES delivery methods. It is established that low electric fields (either continuous or periodic) alter cell transmembrane potential, which leads to the opening of voltage-gated ion channels or changes in the enzymatic activity of phosphatases, containing a voltage-sensor domain [66]. High electric fields might have thermal effects and cause tissue damage; however, when high-intensity stimulation is applied for short intervals, it can trigger positive effects, similar to those induced by longer low electric field strength stimulation [67]. Depending on the pulsed electric field strength, the duration and the number of pulses can be reduced, providing flexibility in the development of parametric protocols (Table 2) [68]. For example, nanosecond PEFs (nsPEFs) feature an extremely high electric field strength (i.e., up 100 kV/cm), which can be induced in 1 to a few hundred nanoseconds. Microsecond pulses can cause reversible electroporation with a field strength between 400 and 600 V/cm, while irreversible electroporation of neurons and cardiomyocytes is caused by electric field of 1–2 kV/cm with at least 30–50 pulses [69,70]. Exceeding the electroporation threshold causes the development of hydrophilic pores in the cell membrane, which leads to increased concentration of intracellular Ca^2+^, altered proliferation, differentiation or even cell apoptosis [71,72]. Therefore, the duration of cell membrane polarization plays an important role in the dynamics of electroporation. For example, in the supra-electroporation range (where pulse duration is shorter than the duration of membrane polarization), PEF intensities of 2–10 kV/cm can still induce reversible damage to the cell. Nevertheless, an increase in the number of pulses (e.g., 10 kV/cm × 900 ns × 600 pulses delivered at 2 Hz) can trigger apoptosis [73]. The application of the same protocol with shorter pulses improves cell viability, which is in accordance with established knowledge. Both microsecond and nsPEF can be used for plasma membrane permeabilization [68], which potentially enables mimicking a spontaneous cellular oscillation with a Ca^2+^ spike [74]. The application of extremely short pulses (typically sub-100 ns) also enable selective permeabilization of internal cell membranes, such as the endoplasmic reticulum, mitochondria or nucleus and induces Ca^2+^ release from the endoplasmic reticulum through IP3-dependent Ca^2+^ channels or permeabilize the membrane of the ER [35,72,75,76].

At the same time, stimuli that are below the electroporation threshold can mimic cell-signaling mechanisms, such as Ca^2+^ signaling pathways as well as mitochondria- and caspase-dependent mechanisms [72].

A wide variety of ES protocols with different parameters have been used in studies with chondrocytes or MSCs in vitro causing contrasting effects. Electric fields of different strengths and frequencies have multiple effects in chondrocytes and MSCs (Figure 2). ES of 2–500 mV/mm acting via ion channels (VGCC, P2X4) and enzymes (phospholipase-C (PLC)) causes changes in gene transcription (Figure 4), resulting in increased cell proliferation, directional migration and differentiation, while nsPEF modulates signaling pathways (Wnt/β-catenin, JNK/CREB-STAT3, ERK/JNK/STAT3) and also affects mRNA expression [16,44,78]. The effects that ES have on chondrocytes and chondrogenesis are summarized below.

Proliferation and metabolic activity. Proliferation of chondrocytes is necessary for cartilage repair [79]. A total of 100 mVRMS (corresponds to 5.2 × 10^−5^ mV/cm) electric fields increased the production of ECM components (collagen type II, glycosaminoglycans (GAGs)) and proliferation in chondrocytes isolated from post-traumatic, non-OA human cartilage [45]. The opposite results were obtained using nsPEF—the proliferation of porcine chondrocytes was increased, but dedifferentiation of chondrocytes was also enhanced without showing any cytotoxicity or signs of changing cell morphology. This effect was reduced by inhibiting the wnt/β-catenin pathway [36]. Krueger et al. also evaluated that together with increased proliferation low-intensity (100 mVRMS (corresponds to 5.2 × 10^−5^ mV/cm), ES decreased the metabolic activity of chondrocytes, isolated from osteoarthritic and non-degenerative chondrocytes [45], while alternating electric field (1 kHz, 0.7 VRMS) did not change the metabolic activity of human chondrocytes or bone marrow MSCs (BMMSCs) but increased expression of chondrogenic genes (collagen type II, aggrecan) [80].

Migration. There is much data about the stimulatory effect of ES to cell directional migration due to the negative cell surface potential [1], which was also shown in chondrocytes. Direct current of an at least 80 mV/mm electric field for 2 h induced the inositol phospholipid pathway-dependent cathodal migration of primary bovine chondrocytes, cultivated in a monolayer [78]. Physiologically relevant ES stimulates phospholipase C (PLC)-coupled surface receptors; PLC hydrolyses phosphatidylinositol 4,5-bisphosphate into inositol 1,4,5-trisphosphate (IP_3_) and diacylglycerol (DAG). IP_3_ induces iCa^2+^ release from the endoplasmic reticulum and activates gene expression [81,82]. ES also changed the morphology and alignment of chondrocytes—it caused cell elongation and perpendicular alignment to minimize the electric field gradient across the cell [78].

ATP Oscillations. Ca^2+^-driven ATP oscillations via P2X4 receptors and cAMP/PKA signaling are necessary for prechondrogenic condensation [43]. Electrical stimulation of 500 mV/mm, 10 Hz, 8 ms was observed to cause ATP oscillations via the P2X4 receptor, leading to mouse MSC condensation and increased expression of chondrogenic markers during chondrogenic differentiation [16].

Condensation. Condensation is the initial step of chondrogenic differentiation when a microenvironment for cell differentiation is created [83]. ES of 500 mV/mm, 10 Hz, 8 ms for 3 days induced prechondrogenic condensation in micromass cultures of mouse MSCs and human dermal fibroblasts [16,22]. ES induces prechondrogenic condensation via TGF-β signaling and Ca^2+^/ATP oscillations, which can be inhibited by the paracrine factor secretion inhibitor BFA and gap junction inhibitor carbenoxolone, while bone morphogenetic protein-2 (BMP-2) signaling inhibitor noggin did not exhibit any effect [16].

Secretion of growth factors. Many growth factors play a significant role in chondrogenic differentiation [84]. Some studies showed increased gene expression and secretion of TGF-β1, a key factor for chondrogenic differentiation in vitro, in canine adipose-derived MSCs (ADSCs) mouse MSCs and human fibroblasts after 3 days of ES [16,20,22]. ES of 500 mV/mm, 10 Hz, 8 ms also elevated secretion of PDGF-AA, IGFBP-2 and 3 in human dermal fibroblasts during chondrogenic differentiation [22], and gene expression of BMP2 in mouse MSCs [16] (Table 3).

Production of ECM and MMPs. ECM creates a microenvironment that facilitates chondrocyte attachment and induces chondrogenic differentiation. Components of ECM (collagen II, aggrecan, GAGs) are widely analyzed as biomarkers of chondrogenic differentiation [93]. ES can affect the production of ECM in two ways—by changing the expression of ECM components or by affecting the degradation of ECM by MMPs. Low-intensity electric field stimulation (100 mVRMS (corresponds to 5.2 × 10^−5^ mV/cm), 1 kHz) increased gene expression of aggrecan (ACAN) and GAGs in human chondrocytes, seeded on collagen elastin scaffolds, in comparison to unstimulated control [45]. An electrical field of 500 mV/mm, 10 Hz, 8 ms increased gene expression of chondrogenic markers (collagen II, aggrecan, SOX9), increased protein expression of collagen II, increased GAGs content and decreased gene and protein expression of hypertrophic marker collagen I in murine MSCs and human dermal fibroblasts [22]. Optimized conditions of coupled electrical field (60 kHz, 2 mV/mm cyclic stimulation) when applied for 1 h increased aggrecan mRNA expression, and for 6 h—collagen type II; 30 min continuous stimulation reduced expression of MMP-1, MMP-3, MMP-13 and ADAMTS-4, and ADAMTS-5. These effects can be completely blocked by inhibitors of VGCCs, calmodulin activation, calcineurin activity, phospholipase C activity and prostaglandin PGE2 synthesis. These results show that the effects of cyclic stimulation act mostly via Ca^2+^ signaling [44]. ES also has different effects in hypoxic and normoxic conditions. While ES increased collagen type II and aggrecan mRNA expression in BMMSCs in both conditions, ES increased collagen type II synthesis in chondrocytes only in hypoxic conditions [80]. The optimized conditions of nsPEF increased gene expression levels of collagen type II gene, SOX9, and ACAN via the JNK/CREB-STAT3 signaling pathway [35].

De-differentiation and hypertrophy. In some studies, the positive effect of ES is observed along with increased expression of hypertrophic genes. Low-intensity ES increased the expression of COL1 and alkaline phosphatase (ALP) and protein expression of collagen type I [45]; an alternating electric field (700 mVRMS, 1 kHz) also increased the expression of COL1 [80]. Nanosecond-pulsed ES had the strongest hypertrophic effect; 5 pulses of 100 ns at 10 kV/cm or 20 kV/cm downregulated collagen II and SOX9 gene expression and increased expression of collagen I and collagen X mRNA in chondrocytes, but this effect can be partially decreased by blocking the wnt/β-catenin pathway [36]. Another study showed that 60 ns pulse of 5–20 kV/cm increased collagen type X and collagen type I expressions [35].

Thus, the application of ES might result in a range of cellular responses, leading to diverse functional outcomes depending on the protocol and ES type used. ES parameters should be equilibrated based on the utilized ES methods and specific investigated cell types in order to optimize the production of engineered tissue.

## 5. The Effects of ES on Chondrogenesis In Vitro

Current in vitro studies using different ES protocols have generated controversial results on the effect of ES to chondrogenesis (Table 4).

Continuous and cyclic low-voltage ES with optimized conditions can be a valuable tool for the induction of ATP oscillations, resulting in the condensation of cells and increased expression of chondrogenic genes (COL1, ACAN, SOX9) in HDFs, murine BMMSCs and chondrocytes [16,22]. Wang et al. [95] showed that the highest expression of different chondrogenic genes can be achieved in different time points, so each case needs optimization research. High-intensity and low-voltage electrical field stimulation also had a chondroprotective effect because of the reduced expression of interleukin 1β (IL-1β), which induced MMPs.

NsPEF causes mixed effects on chondrogenesis—a shorter pulse (10 ns at 2 × 10^6^ mV/mm, 60 ns at 5 and 2 × 10^6^ mV/mm, 100 ns at 1 × 10^6^ mV/mm) increased the expression of chondrogenic genes in porcine chondrocytes, while a longer pulse (60 ns) with EF of 0.5–2 × 10^6^ mV/mm) mostly increased de-differentiation and hypertrophy of porcine MSC and chondrocytes [35,36]. EF of 2 and 3 × 10^6^ mV/mm with longer pulse duration (100, 300 ns) also decreased the viability of porcine MSCs while similar conditions increased the proliferation of porcine chondrocytes [35]. Mixed results can be explained by the different size of MSC and chondrocytes, which reacts differently to electrical stimuli; also, different signaling pathways are activated by diverse electrical fields.

The mechanism of ES on chondrogenesis is not well understood yet. It is known that ES directly affects membrane channels (e.g., VGCC) and receptors (e.g., P2X4), affecting cAMP/PKA signaling. The studies also show that the inhibition of some signaling pathways (JNK/CREB-STAT3, wnt/β-catenin) decreases or diminishes the effect of ES [43].

Until now, only a few studies have focused on the effects of ES on chondrogenesis in 3D structures (Table 5). Biocompatible and electroconductive scaffolds must be used for such studies. ES was mostly applied on hydrogels or scaffolds containing natural cartilage ECM components (collagen, elastin, or hyaluronic acid) [45,101,102]; however, some studies have been performed with synthetic scaffolds [103]. Both biological and synthetic scaffolds have their own advantages, as natural polymer structures are more biocompatible while synthetic counterparts are mechanically stronger and easier to produce and manipulate [38], which makes it possible to improve conductive properties of the scaffold by, for example, incorporating conductive materials. Moreover, different scaffold manufacturing methods such as 3D printing and electrospinning result in scaffolds exhibiting varying properties (porosity, cell adhesion, etc.) that might affect conductivity [104]. Another possible future prospect of cartilage tissue engineering research is the development of electroactive hydrogels that can be shaped by applying noncytotoxic voltages [105].

In addition to classic applications of ES, pulsed supra-physiological electrical field ES can also lead to increased chondrogenic (collagen, aggrecan) gene and protein expression when used on different origin chondrocytes and MSCs, making it a valuable tool for cartilage tissue engineering.

## 6. Recommendations on Reporting ES in the Context of Chondrogenesis

In order to improve the reproducibility of the results and ensure adequate consolidation of knowledge in the area of ES, pulse parameters should be properly characterized (Figure 5). One of the main parameters is the electric field intensity, which is characterized by electric field strength, pulse-waveform, pulse duration, number of pulses and the repetition frequency. However, other influencing factors are often overlooked. Since ES is cell membrane polarization-based phenomenon, the dynamics of the cell membrane polarization are influenced by the dielectric parameters of cell and the surrounding medium and the frequency of the externally applied electric field [66]. Therefore, comparison of the induced biological effects is non-straightforward and often impossible if the pulsing conditions are not properly reported. One of the solutions is to use the guidelines for reporting data on pulsed electric field-based treatments [77,107,108]. Additionally, simulation of the electric field and the field homogeneity should be reported, since they strongly depend on the structure of the applicators (i.e., electrodes) [109]. Finally, the effects of Joule heating [110,111] should be estimated in case of long-term treatments; thus, the input energy of the whole protocol should be presented.

To exclude possible cytotoxic effects, it is necessary to prove cell viability and proliferative activity after ES, which can be performed via fluorescent imaging (Live/Dead, Calcein AM, EdU) or metabolic assays (CCK-8, AlamarBlue etc.) [112]. The direct effect of electrical stimulation on the cell membrane should be monitored using VGCC inhibitors, while the creation of micropores can be proved via the Yo-Pro-1 uptake [113]. During chondrogenesis assays, it is necessary to not only show chondrogenic gene (SOX9, Col II, Aggrecan), but also hypertrophic gene (Col X, Col I, MMP-13) expression [36]. To prove successful chondrogenic differentiation, tissue GAG expression can be evaluated using histology (AlcianBlue or Safranin-O) or its release quantified in the cell supernatant [45,114]. In addition to the markers of chondrogenic differentiation, signs of undesired effects must also be analyzed. Cytokines such as IL-6 can be monitored as signs of inflammation [115], while tissue degradation can be detected via the expression of proteases such as MMP-13, ADAMTS or the downregulation of previously mentioned chondrogenic genes and proteins [116].

Another important aspect of studies using ES on engineered cartilage tissue is the selection of scaffolds as they can both affect the strength of ES exerted on the cells as well as influence cell viability and chondrogenesis. Properties such as mechanical strength, conductivity and biocompatibility of the scaffolding material should be evaluated in the context of the experiment [38]. Furthermore, other scaffold parameters, such as porosity, and the introduction of parallel mechanical stimulation could play a role in the outcomes of ES studies.

## 7. Conclusions

Electrostimulation is still a novel technique that has a high potential in tissue engineering. Electrical impulses have various effects on cellular processes, including the opening of voltage-gated membrane channels and signaling pathway activation. Recent studies showed that ES can be used to increase chondrogenic gene and protein expression in 2D and 3D systems, while some conditions can induce cell hypertrophy or cell damage. Thus, ES is a promising tool for chondrogenesis studies and has the potential for cartilage repair.

## Figures and Tables

**Figure 1 bioengineering-10-00454-f001:**
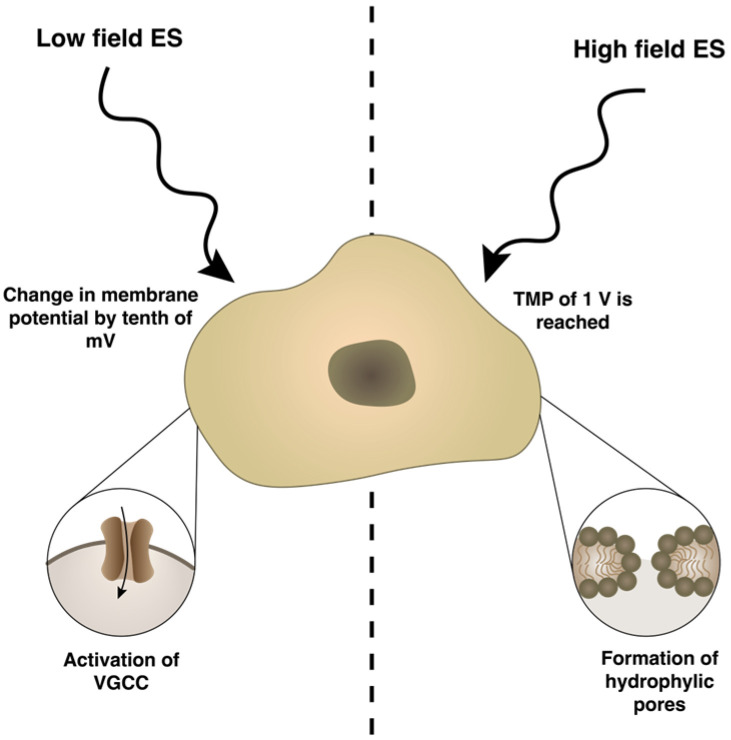
The mechanism of different types of electrical stimulation (ES) on cells membrane. TMP—transmembrane potential, VGCC—voltage-gated calcium channel.

**Figure 2 bioengineering-10-00454-f002:**
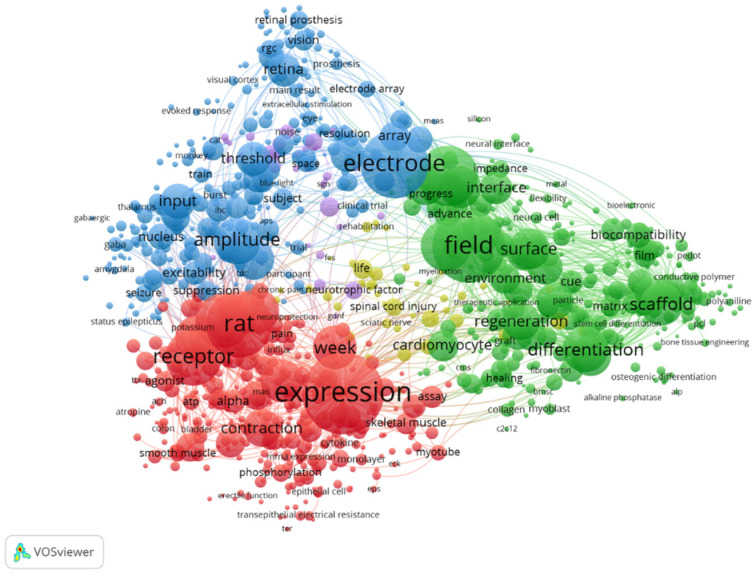
Keyword map of electrical stimulation studies during last 15 years. Visualized using VOSviewer, version 1.6.18.

**Figure 3 bioengineering-10-00454-f003:**
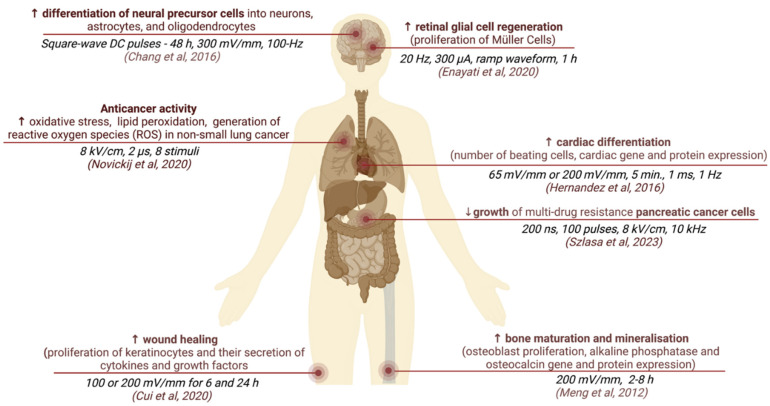
Examples of different ES parameter settings used on different cell types and associated cellular behavior [59,60,61,62,63,64,65].

**Figure 4 bioengineering-10-00454-f004:**
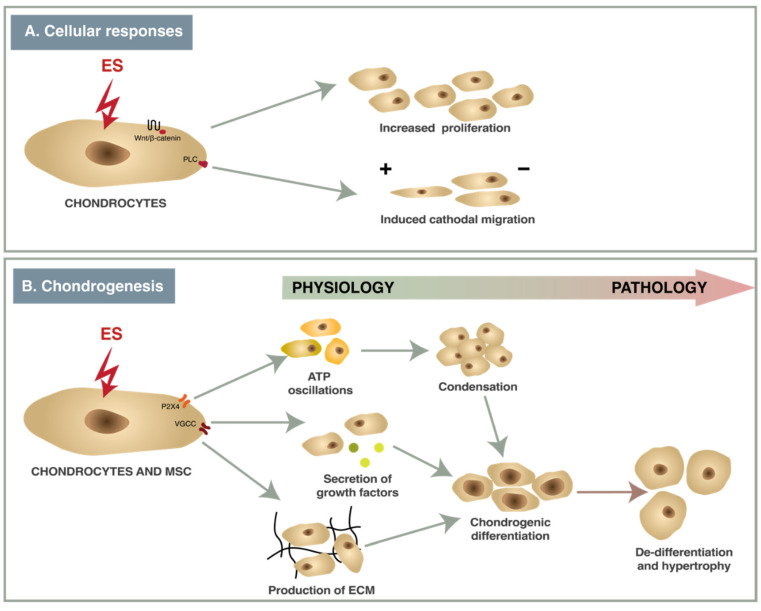
The effects of ES to responses (**A**) and chondrogenesis (**B**).

**Figure 5 bioengineering-10-00454-f005:**
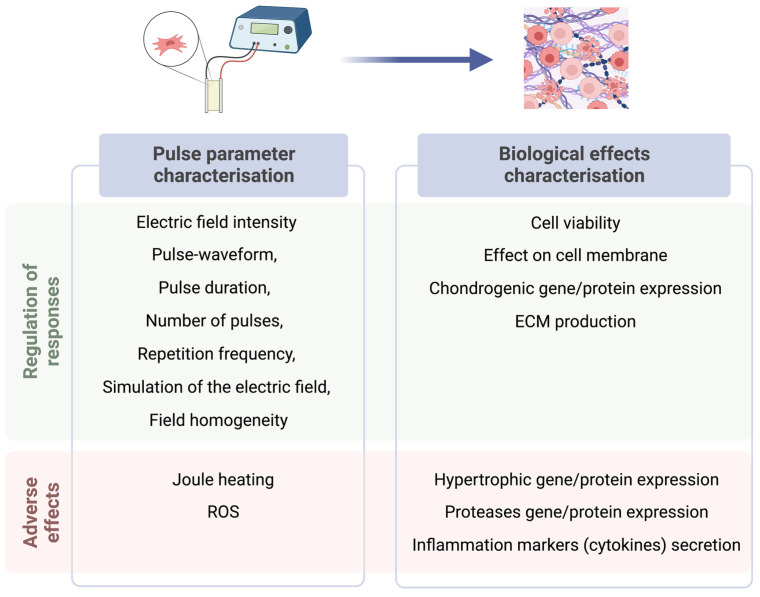
Recommendations on electrical parameters and biological effects that must be characterized in order to get reliable results on chondrogenesis experiments.

**Table 1 bioengineering-10-00454-t001:** The conductivity of cartilage in different models.

Type of Cartilage	Conductivity	Reference
	Baseline—post-exercise	[48]
All cartilages,	1.12–2.98 S/m;	
Patellar cartilage,	1.11–2.80 S/m;	
Trochlear cartilage	1.51–2.98 S/m	
Humeral head bovine articular cartilage *	1.14 ± 0.11 S/m	[49]
Articular cartilage **	0.88 ± 0.08 S/m	[49]

S/m—Siemens per meter. * Results from a 1- to 2-year-old steer. ** Results from a 4-year-old cow.

**Table 2 bioengineering-10-00454-t002:** The effect of different stimulus parameters on cell membranes.

	Cell Type	ES Conditions	Result	Reference
Nanosecond pulse	Cancer cell lines CT-26 and EL-4	300 and 100 pulses (200 ns, 7 kV/cm, 10 Hz)	Induced ER stress	[75]
	Porcine bone marrow-derived stromal cells (pBM-MSCs)	10 ns at 20 kV/cm, 100 ns at 10 kV/cm	Affects intracellular signaling pathways (JNK, P38, ERK, and Wnt signaling pathways)	[35]
	TPC-1 (papillary thyroid carcinoma cell line)	900 ns	Reduced viability and proliferation, induced apoptosis	[73]
Microsecond pulse	Tumor cell lines (DC3F, IGROV 1, SA-1, MCF7, B16F0, TBL.Cl2, TBL.Cl2 PT, HeLa, IGROV 1/DDP, B16F1, MM46T, EAT)	400–600 V/cm, 1 HZ, 100 μs	Reversible plasmic membrane electroporation	[77]
	HL1 cardiomyocytes, PC12, F11, and SH-S5Y5 neural cells	1000–1250 V/cm, 100 μs 30–50 pulses	Irreversible plasmic membrane electroporation	[70]
	Human adipose mesenchymal stem cells (haMSC)	One single micropulse of 100 μs	Induced spontaneous Ca^2+^ oscillations	[74]
	haMSC	One single micropulse of 100 μs	Permeabilization of ER membrane	[76]

**Table 3 bioengineering-10-00454-t003:** The effects of different growth factors on chondrogenesis.

Growth Factor	Function in Chondrogenesis	Reference
Transforming growth factor (TGF)-β1	Induces condensation of MSCs; Induces the production of fibronectin and N-cadherin	[16]
Platelet-derived growth factor (PDGF)-AA	Promotes MSC osteogenic differentiation and migration; Promotes chondrogenesis in the early stages of limb development	[85,86]
Insulin-like growth factor-binding protein (IGFPB)-2	Reduces proliferation of chondrocytes; Stimulates expression of prehypertrophy marker Indian hedgehog; Inhibits chondrogenic differentiation and ECM synthesis of micromass cultures	[87,88]
IGFPB-3	Reduces proliferation of chondrocytes; Might diminish the synthesis of matrix collagen and aggrecan	[89,90]
BMP-2	Increases chondrogenic differentiation by increasing Sox9a and Runx2 proteins expression in vitro, Increases hypertrophy and expression of osteogenic markers: type I collagen, type X collagen	[91,92]

**Table 4 bioengineering-10-00454-t004:** The effects of different types of ES on chondrogenesis.

Cell Type	ES Conditions	Beneficial Effects	Adverse Effects	References
Continuous Stimulation				
Human dermal fibroblasts (HDFs)	100–500 mV/mm electric field, bipolar square-wave pulse, 6–10 ms at 5 Hz for 3 days	8 ms pulse: ↑ condensation; ↑ expression of chondrogenic genes (*COL2A1*, *ACAN*, *SOX9*;) ↓ expression of *COL1A1*, *COL1A2* genes; ↑level of COL2 protein, GAGs; ↑ secretion of growth factors (TGF-β1, PDGF-AA, IGFBP-2 and 3)	Pulse duration 6 ms: No condensation Pulse duration 10 ms: Damaged cells, No compact condensation	[22]
Murine BMMSCs	100–2500 mV/mm electric field, bipolar square-wave pulse, 8 ms at 5 Hz for 3 days	Electrical field of 500 mV/mm: ↑ condensation; ↑ expression of chondrogenic genes (*COL2A1*, *ACAN*, *SOX9*); ↓ expression of *COL1* gene; ↑ level of COL2 protein, GAGs; ↑ expression of TGF-β1 and BMP2.	Electrical field of 100 and 2500 mV/mm: ↓ Ca^2+^/ATP oscillations; ↓ condensation	[16]
Human ADSCs are extracted from subcutaneous abdominal adipose tissue	1 kHz, 20 mv/cm for 20 min.	↑ aggrecan secretion ↑ expression of *COLII* and *SOX9* genes ↓ expression of *COLX* gene		[94]
Bovine chondrocytes	0.02–4 mV/mm, sine-wave with a frequency of 60 kHz; stimulation time 0.5 h	0.5-h of 2 mV/mm (1 min on (1′ ON), 7 off (7′ OFF)—30 cycles and 1′ ON/1′ OFF 30 cycles), harvest after 3.5 h of ES: ↑ expression of *ACAN* gene. 2 mV/mm, regimen—1′ ON/7′ OFF—30 cycles and continuous stimulation, harvest after 5.5 h: ↑ expression of *COL2* gene.	2–6 h of stimulation, 0.1, 0.5, 4 mV/mm amplitude: No change in expression of *ACAN* gene 0.02, 0.1, 0.5, 1, 4 mV/mm amplitude: No change in expression of *COL2* gene	[95]
Explants from human osteoarthritic cartilage	2 mV/mm, static for 30 min followed by 8.33 μs square wave pulse at 60 kHz for 1 h × 4 times a day (gap 5 h) for 7 or 14 days	↑ proteoglycan and collagen content; ↑ expression of *ACAN* and *COL2* genes; ↓ expression of IL-1β induced *MMP-1*, *MMP-3*, *MMP-13*, *ADAM-TS4* genes.	-	[19]
Human ADSCs were extracted from subcutaneous abdominal adipose tissue	Capacitively electric fields (20 mV/cm, 60 KHz) pulsed wave applied for 20 min daily for 7 days.	↑ expression of *COLII*, *SOX9* genes; ↓ expression of *COL X* and *COLI* genes ↑ secretion of aggrecan No difference in cell viability	-	[96]
Articular chondrocytes were isolated from adult bovine patellae	2 mV/mm, 8.33 μs square wave pulse at 60 kHz for 1–6 h 1′ ON/7′ OFF) for 1 h for aggrecan and 1′ ON/1′ OFF for 6 h for collagen II; for MMPs—30 min static stimulation at 2 mV/mm	↑ expression of *ACAN*, *COLII* genes ↓ expression of IL-1β induced *MMP-1*, *MMP-3*, *MMP-13*, *ADAMTS-4*, *ADAMTS-5 genes*	-	[44]
Nanosecond pulsed electrical field				
Normal human chondrocytes (#CC2550, Lonza)	Asymmetrical biphasic rectangular pulses, 210/30 ms in each polarity, respectively, repeating at 4150 Hz, delivered in 10-ms bursts 15 times per second for 30 min. PEF generated peak changes in current of 2.7 mAmps and an electric field in culture media of 0.2 mV/cm	↑ proliferation ↑ NO and cGMP		[97]
Chondrocytes from porcine articular cartilage tissue	1–2 × 10^6^ mV/mm, square wave with transients; 5 × 100 ns, 1 Hz	↑ proliferation of chondrocytes.	↓ GAG production; ↓ expression of *COLII*, *SOX9;* ↑ expression *COLI* gene and *COLX.*	[36]
Porcine BMMSCs	0.5–3 × 10^6^ mV/mm, square wave with transients; 5 × 10^−300^ ns, 1 Hz	10 ns at 2 × 10^6^ mV/mm, 60 ns at 5 and 2 × 10^6^ mV/mm, 100 ns at 1 × 10^6^ mV/mm: ↑ expression of *COLII*, *SOX9*, and *ACAN* genes; 10 ns at × 10^6^ mV/mm and 100 ns at 10 × 10^6^ mV/mm: ↑ production of GAGs	2 and 3 × 10^6^ mV/mm with longer pulse duration (100, 300 ns): ↓ viability; 60 ns at 0.5–2 × 10^6^ mV/mm: ↑ expression of *COLI* and *COLX* genes	[35]
Rat BMMSCs	100 ns duration, 10 kV cm^−1^, 1 Hz)	↑ *OCT4* and *NANOG* expression Together with grelin ↑ expression of *SOX9*, *COLII*, *ACAN* genes ↑ de novo cartilage regeneration (smoother cartilage surface in defect area, ↑ ICRS histology score)	-	[98]
Porcine BMMSCs and human BMMSCs	5 pulses of nsPEFs (10 ns at 20 kV/cm, 60 ns at 5 kV/cm, 60 ns at 10 kV/cm, 60 ns at 20 kV/cm, and 100 ns at 10 kV/cm, 1 Hz) with 1 s time interval between two pulses	10 ns at 20 kV/cm, and 100 ns at 10 kV/cm in both types of cells: ↑ enhance trilineage differentiation potential; -No influence on proliferation; ↑ expression of *OCT4* and *NANOG* genes;	-	[99]
Porcine BMMSCs	Stimulation was carried out with 5 pulses of nsPEFs (10~25 kV/cm, 10~100 ns), and the time interval between each pulse was 1 s.	Four 100 ns at 10 kV/cm pulse on cells cultured on PLLA/CNT films: ↑ tri-lineage differentiation 4 times of PES ↑ expression of pluripotency genes (*OCT4*, *NANOG*, *SOX2*)	Single 100 ns at 10 kV/cm pulse on cells cultured on PLLA/CNT films: ↑ expression of pluripotency genes (*OCT4*, *NANOG*, *SOX2*) 3 days after pulse	[100]

**Table 5 bioengineering-10-00454-t005:** Different effects of ES on cells grown in scaffolds or hydrogels.

Scaffold Characteristics	Cell Type	Electrical Stimulation Parameters	Beneficial Effects	Adverse Effects	Reference
3D collagen I -elastin scaffolds	Human chondrocytes (OA and control)	5.2 × 10^−6^ and 5.2 × 10^−5^ mV/mm, sine wave 1 kHz. 3 times in a day for 45 min for 7 days	5.2 × 10^−6^ mV/mm ↑ GAG, COLII protein synthesis	↑ COLI protein synthesis	[45]
CNT (carbon nanotubes)/PCU polycarbonate urethane	Human chondrocytes (Cell Applications)	Alternating current (AC) stimulation of voltages was generated at 10 lA with 10 Hz throughout the entire cell experiment but with different stimulation times (specifically, either 3 or 6 h)	Both 3 and 6 h stimulation ↑ proliferation of chondrocytes	-	[38]
Collagen I-elastin scaffold	Chondrocytes (non-degraded)	0.005–2.5 mV/mm bipolar wave delivered at 1 or 60 kHz for 45 min per day for seven days.	↑ chondrogenic re-differentiation at the gene and protein level of human de-differentiated chondrocytes	-	[106]
Graphene-Containing PCL/Bioactive Glass Bilayer	Pre-osteoblastic MC3T3-E1 and chondrogenic ATDC5	2 mV/mm, sine wave at 60 kHz 30 min/day for 3 days	↑ viability of ATDC5 cells	↓ viability of MC3T3-E1 cells	[103]
Injectable hyaluronic acid—gelatin hydrogel	Porcine MSCs	0.9–1.2 mV/mm sine wave at 60 kHz, 30 min 4 times a day, 21 days	↑ *SOX9* and aggrecan and COLII protein	↓ production of GAG	[101]

## Data Availability

Not applicable.

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
