# Peer review of "Electrical Stimulation in Cartilage Tissue Engineering"

_bioengineering, 2023, doi:10.3390/bioengineering10040454_

Round 1

Reviewer 1 Report

Please review the full-text(writing and grammar) because there are some sentences and words that have lowered the quality of your text, while the level of the article can be better by re-editing.

Author Response

Dear reviewer,

Thank you, we reviewed the manuscript and corrected the mistakes.

Reviewer 2 Report

The review article by Vaiciuleviciute et al presents an overview of the use of electrostimulation to modulate cell phenotype and an exhaustive review of the application of ES to chondrogenesis.

This review contains a lot of information, but the overall organization of the manuscript could be improved. There are also a number of sentences which should be changed as they do not make sense.

General comments

I do not see any difference between paragraphs 5 and 6. They both address the effects of ES on chondrocytes and chondrogenesis. I think they should be merged.

The title of paragraph 6 mentions “in vitro”, suggesting that there are some ES "in vivo"? In line with this comment, I was wondering whether ES could be used "in vivo" in patients to improve OA symptoms, or whether it is only an "in vitro" technique that favors the production of chondrocytes and a cartilage matrix in tissue engineering approaches.

Minor comments

Paragraph 1, page 1, lines 52-56: It is written that “ES activates a non-specific signal transduction pathway”. This is not a correct statement. Considering the work reported in the following paragraphs, it rather seems that there is not a single non-specific pathway (actually a single non-specific pathway is not a consistent statement), but different signal transduction pathways, all of them affecting Calcium signaling.

Paragraph 1, page 1, lines 58-62: What is the input of the second sentence “additionally, a systemic approach…..protocols”, as it seems that the first sentence already addresses these issues. What is the meaning of the word “systemic” here?

Paragraph 2, page 1, line 67. It is not clear whether ES can lead to both the activation of ion channels and permeabilization of cell membrane, or only one of these processes, depending on the parameters. Please re-write this sentence .

Paragraph 2, page 1, line 73. “result in activation of voltage gated ion channels”. Does this apply to chondrocytes, as I do not think they possess such channels?

Paragraph 2, page 1, line 89. What does "dominating keyword rat" mean?

Paragraph 2, page 2, line 96. Same question as above: are there voltage-gated calcium channels in chondrocytes? Since this is presented as a general mechanism of ES action on cells, it is important to mention whether chondrocytes, or MSCs, express these channels.

Paragraph 2, page 2, line 99-100. The sentence “similarly, intracellular electric signals caused by external mechanical stimulation through mechanotransduction” is not a sentence and does not mean anything as it is.

Paragraph 2, page 2, line 100-101. It is not clear whether mechanical stimulation and electrostimulation have redundant, additive or synergistic effects. This should be clarified.

Paragraph 3, page 4, line 141. It is written “ES studies on cartilage have been carried out in vitro, in silico and in vivo”, but I  couldn’t find studies addressing in vivo studies in the review.

Paragraph 4, page 4, line 146. “from continuous static voltage pulses of various weveforms” the sentence is not complete and should be corrected.

Paragraph 4, page 5, line 160. “the enzyme activity of phosphates containing the voltage-sensor domain” . This part of sentence is very unclear. What is an enzyme activity of phosphates? What is the link between the enzyme activity and the sensor part of the channel. Please clarify.

Paragraph 4, page 5, lines 188-194. This short paragraph introduces the different techniques used to produce ES, and it should be placed at the beginning of section 4.

Figure 3. This figure does not clearly shows the effects of ES. I cannot see which are the positive, stimulatory effects, and the negative, inhibitory actions of ES on the different cellular processes.

Tables 1 and 2. I really appreciated these tables, which provide a lot of useful information.

Paragraph 6, page 11, line 305. “ES directly affects membrane channels”. Is this true for all cells including chondrocytes or MSCs?

Author Response

Dear reviewer,

Thank you very much for valuable comments and suggestions.

Paragraphs 5 and 6 were merged.

Our review was focused on in vitro studies, because most of in vivo studies are related to pain management, which is more rated to neural responses, not cartilage and our focus was on cartilaginous processes.

Paragraph 1, page 1, lines 52-56: It is written that “ES activates a non-specific signal transduction pathway”. This is not a correct statement. Considering the work reported in the following paragraphs, it rather seems that there is not a single non-specific pathway (actually a single non-specific pathway is not a consistent statement), but different signal transduction pathways, all of them affecting Calcium signaling.

Thank you, we corrected it “ES activates several independent signal transduction pathways, therefore it is difficult to establish a direct link between ES and specific cellular responses. It is known, that ES activates JNK/CREB-STAT3, ERK/JNK/STAT3, wnt/β-catenin signaling pathways, leading to enhanced phosphorylation of JNK, CREB, and STAT3 [35] and expression of β-catenin protein [36]“ (the sentence was moved to ...)

Paragraph 1, page 1, lines 58-62: What is the input of the second sentence “additionally, a systemic approach…..protocols”, as it seems that the first sentence already addresses these issues. What is the meaning of the word “systemic” here?

We removed the part of the sentence: “Additionally, a systemic approach and characterization of the...)

Paragraph 2, page 1, line 67. It is not clear whether ES can lead to both the activation of ion channels and permeabilization of cell membrane, or only one of these processes, depending on the parameters. Please re-write this sentence .

ES can lead one process, we rewrote the sentence “…may lead to either activation of ion channels and other voltage sensitive proteins, or permeabilization of the plasma membrane“

Paragraph 2, page 1, line 73. “result in activation of voltage gated ion channels”. Does this apply to chondrocytes, as I do not think they possess such channels?

There is much literature about VGCC in chondrocytes:

            Voltage gated ion channels in rabbit chondrocytes were shown in 1996 (DOI: 10.1016/s0742-8413(96)00091-6). There are reviews about VGCC in chondrocytes (Barrett-Jolley et al, 2010, https://doi.org/10.3389/fphys.2010.00135; Matta et al., 2015 10.1007/s11926-015-0521-4), https://doi.org/10.3390/cells9071577

Paragraph 2, page 1, line 89. What does "dominating keyword rat" mean?

It’s a main in vivo model in osteoarthritis researches

Paragraph 2, page 2, line 96. Same question as above: are there voltage-gated calcium channels in chondrocytes? Since this is presented as a general mechanism of ES action on cells, it is important to mention whether chondrocytes, or MSCs, express these channels.

Yes, they are, the literature about VGCC in chondrocytes is already mentiones, VGCC in MSCs are reviewed here: DOI: 10.1111/cpr.12623

Paragraph 2, page 2, line 99-100. The sentence “similarly, intracellular electric signals caused by external mechanical stimulation through mechanotransduction” is not a sentence and does not mean anything as it is.

Thank you, we corrected it, we rewrote sentence “Furthermore, cytoskeletal structure reorganization, including denser f-actin texture and aligned actin filament orientation has been observed in response to ES [42] as well as inverse when mechanical stimulation causes intracellular electrical signals through mechanotransduction [37].“

Paragraph 2, page 2, line 100-101. It is not clear whether mechanical stimulation and electrostimulation have redundant, additive or synergistic effects. This should be clarified.

We clarified: “The interconnectedness of the effects of ES and mechanical loading might be used in cartilage tissue engineering as scaffolds that cannot withstand mechanical pressure could be instead stimulated with electrical fields causing a similar effect as mechanical load.”

Paragraph 3, page 4, line 141. It is written “ES studies on cartilage have been carried out in vitro, in silico and in vivo”, but I couldn’t find studies addressing in vivo studies in the review.

As we didn’t write about in silico and in vivo studies, we removed this part of the sentence

Paragraph 4, page 4, line 146. “from continuous static voltage pulses of various weveforms” the sentence is not complete and should be corrected.

We corrected the sentence “Such stimulus can be directly applied as straightforward continuous static voltage on tissue culture as well as more complicated stimulation with pulses of various waveforms.“

Paragraph 4, page 5, line 160. “the enzyme activity of phosphates containing the voltage-sensor domain” . This part of sentence is very unclear. What is an enzyme activity of phosphates? What is the link between the enzyme activity and the sensor part of the channel. Please clarify.

We rewrote this part „...or changes in the enzymatic activity of phosphatases, containing a voltage-sensor domain [66].“

Paragraph 4, page 5, lines 188-194. This short paragraph introduces the different techniques used to produce ES, and it should be placed at the beginning of section 4.

Thank you, we moved it.

Figure 3. This figure does not clearly shows the effects of ES. I cannot see which are the positive, stimulatory effects, and the negative, inhibitory actions of ES on the different cellular processes.

We changed the figure

Tables 1 and 2. I really appreciated these tables, which provide a lot of useful information.

Thank you

Paragraph 6, page 11, line 305. “ES directly affects membrane channels”. Is this true for all cells including chondrocytes or MSCs?

Yes, we added the links to articles above

Reviewer 3 Report

The manuscript by Vaiciuleviciute et. al summarizes the application of electrical stimulation in cartilage regeneration. Although the authors explained different manuscripts related to the topic, it is strongly recommended to reorganize the information and include more manuscripts as well as a more profound explanation of the results that are included.   

Comments that need major revisions

1. In the introduction, list some of the diseases that are treated with electrical stimulation.

2. Explain the concept of non-excitable cell types.

3. In the introduction, explain with more detail the following affirmation, along with the pathways that are mentioned in line 202, 213, 264 and 269.

ES activates a non-specific signal transduction pathway, therefore it is difficult to establish a direct link between ES and cellular effect it creates.

Include the pathways that are activated and how they change in the presence of ES.

4. Include a diagram to explain the concepts described in the Electrical stimulation overview section.

5. Explain in more detail how the ES activate ion channels or other voltage sensitive proteins, and how they are related to cartilage regeneration.

6. Include a more detailed explanation of the process electroendocytosis

7. Explain with more detail how cytoskeletal structure reorganization has been observed in response to ES.

8. Include a section to explain the electrical properties of cartilage and how exhibits electromechanical, depth-dependent properties.

9. Include a table that summarizes the experiments that are explained in the paragraph that starts in line 157.

10. Improve presentation of figure 3 and reorganize its information towards chondrogenic differentiation.

11. Explain the meaning of non-degenerative human cartilage

12. Include an explanation of the inositol phospholipid pathway and how decrease the membrane surface potential

13. Include a summary table of the studies mentioned on line 240, along with the function of each transcription factor involved in the chondrogenic differentiation

14. How is the behavior of the cells described in line 254 without ES.

15. Explain possible reasons of the results explained in the section De-differentiation and hypertrophy as well as lines 296-303

16. The manuscript should be checked carefully for grammar. Numerous grammatical errors were identified.

17. Continuity in the manuscript could be improved. It is recommended to use more connectors and finish the paragraphs with a conclusion of the discussion that was described.

Author Response

Dear reviewer,

Thank you for your valuable comments and suggestions. Our response:

Comments that need major revisions

  1. In the introduction, list some of the diseases that are treated with electrical stimulation.

Thank you, we added „...such as movement, psychiatric and seizure disorders“

  1. Explain the concept of non-excitable cell types.

„Due to a lack of voltage gated Na+ and Ca2+ channels these non-excitable cells cannot generate action potential as a response to membrane depolarization“

  1. In the introduction, explain with more detail the following affirmation, along with the pathways that are mentioned in line 202, 213, 264 and 269.

ES activates a non-specific signal transduction pathway, therefore it is difficult to establish a direct link between ES and cellular effect it creates.

Include the pathways that are activated and how they change in the presence of ES.

We added infromation (this is in part 2. Electrical stimulation overview) : „ES activates several independent signal transduction pathways, therefore it is difficult to establish a direct link between ES and specific cellular responses. It is known, that ES activates JNK/CREB-STAT3, ERK/JNK/STAT3, wnt/β-catenin signaling pathways, leading to enhanced phosphorylation of JNK, CREB, and STAT3 [35] and expression of β-catenin protein [36].“

  1. Include a diagram to explain the concepts described in the Electrical stimulation overview section.

Included

  1. Explain in more detail how the ES activate ion channels or other voltage sensitive proteins, and how they are related to cartilage regeneration.

Included „Typically, low electric fields are used in applying ES, which leads to moderate change in membrane potential by a tenth of mV and initiates movement of voltage sensing domains resulting in conformational changes and the opening of voltage gated ion channels (Fig.1). The most important voltage gated channel for chondrogenic differentiation is L-type voltage gated calcium channel (VGCC), which regulates expression of chondrogenesis markers (SOX9, COL2A1, Ihh) in vitro and limb development in vivo [23].“

  1. Include a more detailed explanation of the process electroendocytosis

We added „...which means enhanced absorption of macromolecules after cell exposure to low electric fields“

  1. Explain with more detail how cytoskeletal structure reorganization has been observed in response to ES.

Included:“ Furthermore, cytoskeletal structure reorganization, including denser f-actin texture and aligned actin filament orientation has been observed in response to ES [42] as well as inverse when mechanical stimulation causes intracellular electrical signals through mechanotransduction [37].“

  1. Include a section to explain the electrical properties of cartilage and how exhibits electromechanical, depth-dependent properties.

Included „Cartilage is a biphasic tissue, composed of solid phase, consisted of charged porous collagen-proteoglycan matrix and interstitial fluid phase [46]. Changes in cartilage composition and arrangement of collagen fibers result in different biomechanical prop-erties. It was shown that resistivity gradually increases from superficial zone of cartilage to deep, which means that superficial zone of cartilage contains more mobile charged particles that deeper zones and conducts electrical charge more efficiently. The elastic modulus also increases going from superficial zone to deep while permeability of cartilage decreases [47]. There is not much data about the conductivity of articular cartilage (Table 1). More studies in this field are needed because part of the data was obtained using animal models, which do not exactly reflect human data.“

  1. Include a table that summarizes the experiments that are explained in the paragraph that starts in line 157.

Included

  1. Improve presentation of figure 3 and reorganize its information towards chondrogenic differentiation.

The figure was changed

  1. Explain the meaning of non-degenerative human cartilage

We added: “post-traumatic, non-OA human cartilage”

  1. Include an explanation of the inositol phospholipid pathway and how decrease the membrane surface potential

Direct current of at least 80 mV/mm electric field for 2 hours induced inositol phospholipid pathway dependent cathodal migration of primary bovine chondrocytes, cultivated in a monolayer [78]. Physiologically relevant ES stimulates phospholipase C (PLC)-coupled surface receptors, PLC hydrolyses phosphatidylinositol 4,5-bisphosphate into inositol 1,4,5-trisphosphate (IP3) and diacylglycerol (DAG). IP3 induces iCa2+ release from endoplasmic reticulum and activates gene expression [81,82]. ES also changed morphology and alignment of chondrocytes – it caused cell elongation and perpendicular alignment to minimize the electric field gradient across the cell [78].

  1. Include a summary table of the studies mentioned on line 240, along with the function of each transcription factor involved in the chondrogenic differentiation

Included

  1. How is the behavior of the cells described in line 254 without ES.

We are not sure that we understand the question correclty, but we added: „Low intensity electric field stimulation (100 mVRMS (corresponds to 5.2 × 10-5 mV/cm), 1kHz) increased gene expression of aggrecan (ACAN) and GAGs in human chondrocytes, seeded on collagen elastin scaffolds, in comparison to unstimulated control“

  1. Explain possible reasons of the results explained in the section De-differentiation and hypertrophy as well as lines 296-303

We added „Mixed results can be explained by different size of MSC and chondrocytes, which reacts differently to electrical stimuli, also different signaling pathways, activated by diverse electrical fields.“

  1. The manuscript should be checked carefully for grammar. Numerous grammatical errors were identified.

Thank you, we checked the mistakes

  1. Continuity in the manuscript could be improved. It is recommended to use more connectors and finish the paragraphs with a conclusion of the discussion that was described.

We added conclusions under paragraphs

Round 2

Reviewer 1 Report

Dear Authors,

Thank you for editing the manuscript .

All the best.

Reviewer 3 Report

The authors have addressed all the comments.